# Tracking Amorphous Calcium Carbonate Crystallization Products with Far-Infrared Spectroscopy

**Boyang Gao** [1] **and Kristin M. Poduska** [1,2,*]

[1] Department of Chemistry, Memorial University of Newfoundland & Labrador, St. John's, NL A1C 5S7, Canada

[2] Department of Physics & Physical Oceanography, Memorial University of Newfoundland & Labrador, St. John's, NL A1B 3X7, Canada

[*] Correspondence: kris@mun.ca

**Abstract:** We prepared solution-precipitated amorphous calcium carbonate by two similar methods and tracked structural changes over time as they crystallized. By cross-referencing mid-range infrared (400–4000 cm$^{-1}$) with far-infrared (100–400 cm$^{-1}$) spectral features, and by comparing with powder X-ray diffraction data for the aged crystallized products, we provide guidelines for—and potential limitations of—using far-infrared spectroscopy to assess multi-phase Mg-containing calcium carbonate samples that include amorphous or poorly crystallized components.

**Keywords:** infrared spectroscopy; amorphous calcium carbonate; aragonite; dolomite; hydromagnesite





## 1. Introduction

Phase control of calcium carbonate, including its amorphous form (ACC), has been an active area of research over the past few decades. Understanding ACC crystallization processes is a strong focus in the field of biomineralization [1–5]. Amorphous phases are also gaining more attention related to geogenic minerals, including mineral exploration [6], dolomite formation pathways (related to fundamental questions about ocean chemistry) [7], and calcium carbonate deposits at sites of serpentinization reactions (related to CO$_2$ capture and storage) [8]. In technological applications, ACC is being considered to develop effective environmental remediation strategies (for removing ionic contaminants) [9] and as a component in less CO$_2$-intensive binder materials in cement and concrete [10].

ACC crystallization can happen very rapidly, often in a time frame of minutes to hours. As a result, there is extensive literature on synthesis strategies that are effective for stabilizing ACC so that it can be studied more effectively. For example, recent work [7] provides a detailed study that compares how different background ions can influence the dehydration process that can trigger carbonate mineral formation in aqueous solution.

In general, amorphous phases are challenging to identify and track, since the lack of long-range periodicity precludes the effective use of diffraction-based techniques throughout the crystallization process. In order to identify amorphous calcium carbonate phases—and to track their crystallization—it is mid-infrared (MIR) spectroscopy (in the 400–4000 cm$^{-1}$ range) that has emerged as a benchmark tool [11] because of its wide availability and ease of use. Raman spectroscopy is also useful and widely available [12,13], but interference from fluorescence can be a serious obstacle when analyzing Fe-bearing or biological samples. Other techniques that provide valuable structural information for amorphous phases, such as solid-state NMR, X-ray absorption spectroscopy, or X-ray scattering, are used more sparingly due to the considerable infrastructure and time requirements [14–17].

While mid-infrared spectra, which target vibrations within a single carbonate moiety, are common and easy to access, far-infrared spectral analyses, which access lower-energy lattice vibrations, are rare. This is due in part to instrument requirements: the KBr-based

optical components used in the mid-IR range are strongly absorbing in the FIR, and acquiring spectra under vacuum is essential to remove background water features that permeate the FIR range [18]. Alternative techniques that access the same energy ranges, such as THz spectroscopy, are not routinely applied to carbonates. However, there are useful and detailed studies that catalog and explain differences in THz [19] and FIR spectra [20] related to crystalline single-phase carbonate minerals. Attention to ACC in FIR is even more limited; a recent work [21] relies on comparisons of MIR and FIR spectra of lab-produced ACC with biogenic ACC produced by cyanobacteria.

In this work, we apply FIR spectral interpretation to ACC samples by comparing their spectra while they are fresh (amorphous) and after sufficient periods of time that they have begun to crystallize. This builds on earlier work by our group [18] and others [20–22] that develops a framework for linking FIR and MIR spectra. Here, we extend this framework to focus on more challenging—and more realistic—cases of poorly crystalline and multi-phase calcium carbonate samples.

## 2. Materials and Method

### 2.1. Synthesis

For this spectroscopy-focused study, we needed ACC specimens that crystallized slowly enough to track over a span of days to months. To achieve this, we modified existing recipes in the literature, using Mg salts as well as ethanol (EtOH) in an aqueous precipitation synthesis. Others have shown that using high Mg:Ca ratios helps to stabilize amorphous calcium carbonate [23,24]. Others [25] report that EtOH can promote amorphous calcium carbonate formation, and that EtOH:water composition during synthesis has an impact on how the amorphous products eventually crystallize [26].

Our synthesis used Ca:Mg at 1:5, starting with a stock solution that contained both $CaCl_2$ (ACS reagent grade (Sigma Aldrich, Oakville, ON, Canada) and $MgCl_2$ (ACS reagent grade, EMD Chemicals, Oakville, ON, Canada). In total, there was 10 mL (Ca at 60 mL and Mg at 300 mM), to which $Na_2CO_3$ (ACS reagent grade (Sigma Aldrich), 10 mL at 60 mM) was then added. All solutions were held at 10 °C in an ice bath before mixing. In some experiments, EtOH was added at the same time as the $Na_2CO_3$ solution (20 mL of EtOH to give 1:1 volume ratio with the total water). Precipitation occurred immediately. Centrifugation separated the supernatant from the precipitates, which were then filtered and dried at room temperature for 12 h. For storage, the dried solids were sealed in 5 mL Eppendorf tubes for up to 2 years. Periodically (days to months), small amounts of powder were withdrawn for structural characterization measurements.

### 2.2. Characterization

All FTIR data were collected with an attenuated total reflection (ATR) geometry using Bruker spectrometers (Billerica, MA, USA): Alpha (for mid-IR) or Vertex 70 V system (for mid-IR and far-IR). In both cases, the ATR accessory used a diamond window with a single bounce at 45 degrees (Bruker Platinum ATR). Spectra were compared with those from an existing database focusing on biominerals and archaeologically relevant specimens [27].

Powder X-ray diffraction (PXRD) data were collected at room temperature in a transmission geometry with a glass capillary using a Rigaku (The Woodlands, TX, USA) XtaLAB Synergy-S X-ray diffractometer using a Cu source. Data collection and data processing, including lattice constant refinements, used the CrystAlisPro software in conjunction with the International Crystal Structure Database (ICSD, Karlsruhe, Germany) for phase identification [28].

## 3. Results

### 3.1. Mid-IR (MIR) Spectra

Representative MIR ATR-FTIR spectra for the synthesized product, with and without EtOH addition, are shown in Figure 1. The fresh samples (Figure 1a,b,e,f) show peak profiles that are characteristic of ACC, with distinct in-plane carbonate stretching ($\nu_3$, near

1500 cm$^{-1}$) and out-of-plane carbonate stretching ($\nu_2$, near 875 cm$^{-1}$), small symmetric stretch ($\nu_1$, near 1090 cm$^{-1}$), and a very broad hump corresponding to the in-plane carbonate wag ($\nu_4$, near 700 cm$^{-1}$) [27,29]. In addition to the carbonate-related peaks, there are water-related (OH) features at 3500 and 1600 cm$^{-1}$. These water-related peaks begin to decrease within days after synthesis; they decrease more after heating to 100 °C for multiple hours but do not disappear completely. (Supplementary Material Figure S1 shows representative examples of the corresponding MIR ATR-FTIR spectra after heating.)

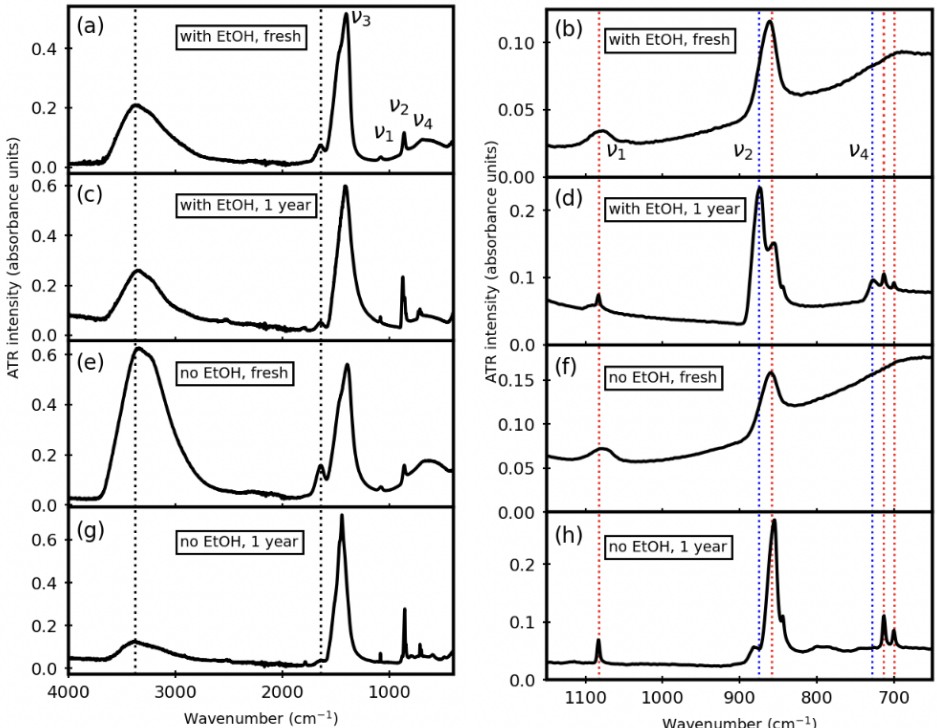

**Figure 1.** Representative ATR-FTIR spectra in the mid-IR range (400–4000 cm$^{-1}$) for synthesized ACC (**a**,**b**) with EtOH (fresh), (**c**,**d**) with EtOH (aged 1 yr), (**e**,**f**) without EtOH (fresh), and (**g**,**h**) without EtOH (aged 1 yr). The black dotted lines correspond to spectral features related to water (OH). The zoomed plots (**b**,**d**,**f**,**h**) emphasize the $\nu_1$ region (1050–1100 cm$^{-1}$), $\nu_2$ region (800–920 cm$^{-1}$), and $\nu_4$ region (650–800 cm$^{-1}$). The red dotted lines indicate aragonite peak positions, and the blue dotted lines indicate dolomite peak positions.

The carbonate-related spectral features change as the samples age, and Figure 1 illustrates that there are considerable differences between the EtOH and no-EtOH preparations. In brief, aging leads to more crystalline products that include multiple phases. (Supplementary Material Figure S1 shows representative examples of the corresponding ATR-FTIR spectra after aging for a broader range of time, from 3 days to 2 years.)

With EtOH (Figure 1a,c), dolomite crystallizes as the dominant phase, with aragonite as a minority phase. The most informative spectral regions for making these phase determination are shown in the zoomed-in region in Figure 1b,d: the $\nu_2$ region (800–950 cm$^{-1}$) and near the $\nu_4$ region (600–800 cm$^{-1}$) [27,29]. In the case of our aged with-EtOH sample, we find that the dolomite $\nu_4$ peak near 728 cm$^{-1}$ is very broad relative to the sharp aragonite $\nu_4$ peaks at 700 and 712 cm$^{-1}$.

Without EtOH (Figure 1c,d), aragonite crystallizes as the dominant phase: there is a strong $\nu_2$ peak at 858 cm$^{-1}$, a small sharp peak at 1083 cm$^{-1}$, and doublet $\nu_4$ peaks at 700 and 712 cm$^{-1}$ [27]. There is also at least one other minority phase, showing a shoulder on the $\nu_2$ peak near 885 cm$^{-1}$.

### 3.2. Powder X-ray Diffraction (PXRD) Data

Since the aged samples show crystalline multi-phase material, we used powder X-ray diffraction (PXRD) to confirm the phase assignments. We note that attempts to do this on the fresh sample were predictably unsuccessful due to their poor crystallinity.

Figure 2 shows which peaks were associated with the dominant crystallized phases. In each case, lattice constant refinements were possible with only one phase. For the sample prepared with EtOH (Figure 2a), dolomite was the dominant phase: 10 peaks can be indexed to an hexagonal unit cell with $a$ = 4.85(1) Å, $c$ = 16.22(5) Å, compared with ICSD #40971 for dolomite ($a$ = 4.823 Å, $c$ = 16.227 Å). For the sample prepared with no EtOH (Figure 2b), aragonite was dominant: 13 peaks can be indexed to an orthorhombic unit cell with $a$ = 4.968(7) Å, $b$ = 7.98(1) Å, $c$ = 5.744(7) Å, which compares well with ICSD #34308 for aragonite ($a$ = 4.960 Å, $b$ = 7.964 Å, $c$ = 5.738 Å).

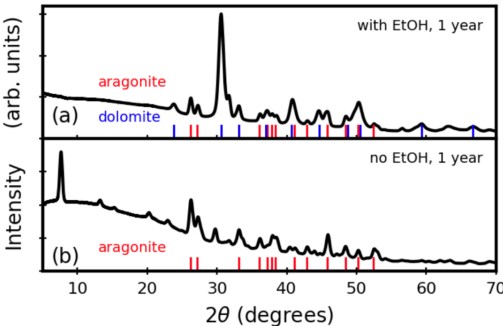

**Figure 2.** Representative PXRD data for samples after 1 year of aging: (**a**) with EtOH; and (**b**) without EtOH. The red lines correspond to peaks that were indexed to aragonite; the blue lines correspond to peaks indexed to dolomite.

The minority phases could not be indexed well due to the fact that many peaks had low intensities and overlapped with nearby peaks. By comparing with data from ICSD standards, we find that with EtOH (Figure 2a), aragonite (ICSD #3408) is an excellent match for the secondary phase, accounting for all non-dolomite diffraction peaks. However, it is more complicated to assess the secondary phase(s) in the aged no-EtOH sample (Figure 2b). Our attempts to assign all peaks to one or two additional phases were not conclusive. The closest matches were magnesium calcite phases (such as $Mg_{0.1}Ca_{0.9}CO_3$, ICSD #10405), as well as hydromagnesite phases (such as $Mg_5(CO_3)_4OH_2·5H_2O$, ICSD PDF #29-0858).

### 3.3. Far-IR (FIR) Spectra

Since the PXRD data do not allow definitive identification of the minority phases in the products, we used FIR spectra (100–600 cm$^{-1}$) to help with phase assignments. Figure 3a,c demonstrates that both fresh samples are virtually indistinguishable, showing a broad peak near 300 cm$^{-1}$, with a shoulder near 400 cm$^{-1}$. The broad peak is centered at a higher wavenumber than the main peak in the aragonite standard (Figure 3e). For the aged samples (Figure 3b,d), the FIR spectral peak shapes are much more complex, which is to be expected given that each sample is multi-phase.

FIR spectra have advantages when assessing the more crystallized aged samples. Figure 3a–c have overlaid black dotted lines to indicate the positions of peaks that appear in the aged no-EtOH sample that do not appear in the standards or the with-EtOH aged sample. By comparing these spectral features with spectra from other works [20], we find a good match for these five peaks with hydromagnesite phases. These phases can incorporate varying amounts of OH and structural water, which leads to a rich array of possible structures. While this structural diversity creates challenges for diffraction techniques that rely on long-range periodicity to obtain structural information, IR spectra are better able to assess vibrations in these OH and water constituents even when there is poor long-range periodicity, as there is in amorphous or poorly crystallized solids.

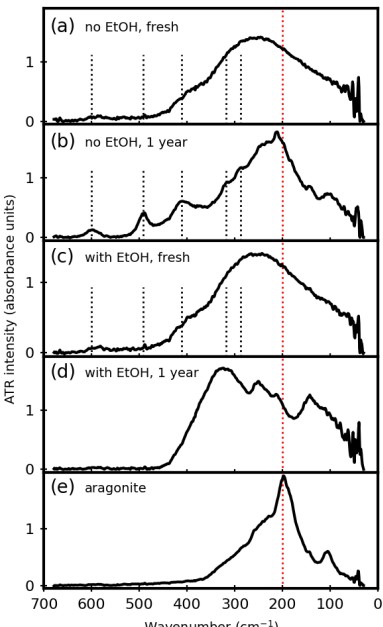

**Figure 3.** Far-IR spectra for samples prepared with and without EtOH (fresh and aged 1 year), along with an aragonite standard. The red dotted vertical line indicates the main aragonite peak. The five black dotted vertical lines indicate peaks that appear only in the no-EtOH sample after aging.

For the aged sample made with EtOH (Figure 3d), definitive peak assignments in the FIR range are not feasible because of the complicated peak shapes. For example, our MIR spectra (Figure 1c,d,g,h) and PXRD data (Figure 2a,b) show that aragonite is present in both aged samples. However, the single-phase aragonite FIR spectrum (Figure 3e) is complicated, as shown in earlier work by others [20]. For single-phase dolomite, this same earlier report [20] provides a detailed description of the FIR spectra for dolomite, which includes six different peaks due to lattice vibrations. Thus, while the FIR spectrum for our sample made with EtOH is consistent with what one would expect for a mixture of dolomite and aragonite, it would be hard to make a solid case for this based on the FIR spectra comparisons alone due to the large number of overlapping peaks.

*3.4. Discussion*

FIR and Amorphous Carbonates

Our ACC samples crystallize over a span of months to produce multi-phase products that include Mg. This diversity of crystalline products is not surprising, based on the literature.

First, it is important to note that ACC itself is not a single unique structure. Others have shown short-range structural differences in ACC can be triggered under different aqueous synthesis conditions. For example, there are reports of pH-dependent ACC structures, such as neutral pH favoring proto-calcite ACC and alkaline pH favoring proto-vaterite ACC [14]. Others [30–32] found that $Mg^{2+}$ can be incorporated into the ACC structure leading to proto-aragonite, perhaps because Mg triggers a more significant energy barrier to pre-calcite cluster nucleation than that of aragonite nucleation, causing preferential aragonite crystallization [33]. More recent studies have followed the crystallization of proto-dolomite [34], as well as the role of Mg with water in triggering crystallization [35]. Focusing on the role of solvent polarity, there is also extensive work to understand mechanisms that control phase selectivity during precipitation from water-EtOH mixtures, many of which involve amorphous precursor stages [36–38]. Interestingly, some results indicate that $Ca^{2+}$ and $CO_3^{2-}$ ions have different solvation tendencies in water-EtOH mixtures, resulting in local inhomogeneity [39], which can in some cases cause a disordered ACC that does not have proto-crystalline local structures [40]. Thus, there is clear evidence from other work

that synthesis conditions can influence structural differences in the amorphous material that dictate which phase(s) will crystallize.

Our MIR and FIR spectra show very few differences between the fresh samples (with and without EtOH), even though their crystallized products are quite different. Like the MIR spectra, the FIR spectra of the fresh ACC do not show definitive differences that would allow us to know whether aragonite or a calcite-type phase (magnesium calcite or dolomite) would be the preferred crystallized phase. There could be two explanations for this. One possibility is that neither MIR or FIR are effective at distinguishing between proto-aragonite, proto-dolomite, and another proto-carbonate structures based on the carbonate-related vibrational modes. The second possibility is that the ACC samples themselves do not have distinctive proto-crystalline structures, such as reported by others using rapid precipitatation from solvents with high alcohol contents [40]; in this case, there would be no reason to expect differences in the MIR and FIR spectra. Our data cannot help us distinguish between these two possible explanations.

Despite the ambiguity, we note that this is an important finding. Other studies in the MIR range [11] have demonstrated that amorphous and crystalline carbonates are easily distinguishable based on the sharpness of the $\nu_4$ peak (corresponding to the in-plane carbonate wag). However, to this time, there are no studies that have been able to demonstrate that more detailed structural information about the amorphous carbonate phases can be extracted from MIR or FIR spectra; the present study does not provide an advance in this way.

We also point out that vaterite would be a possible intermediate phase during the transformation from ACC to either aragonite or calcite. In our data, however, we do no see any evidence of vaterite in the MIR spectra: most notably, there is no peak near 744 cm$^{-1}$ in any of the spectra, which is a peak that vaterite would have [27,29]. In PXRD data, vaterite would also have a distinctive trio of peaks between 20–30$^\circ 2\theta$, which do not appear in any of our samples [28].

### 3.5. FIR and Mg-Containing Phases

Others have conducted thorough comparisons of FIR spectra for many different Mg-containing carbonates [20] (and in the same energy range using THz spectra [19]), and these earlier findings are a great aid for interpreting the data presented herein. More specifically, spectra in these other works help to provide context and backing for the spectral features that we associate with dolomite and the hydrated/hydroxylated phases in our own samples. It is important to note that we do not attempt detailed peak position comparisons; this is because the spectral peak shapes from our attenuated total reflectance (ATR) spectra appear somewhat different from those collected using a transmission geometry [20] or THz-time domain spectroscopy [19]. We emphasized the origin of these spectral differences due to sample measurement geometries in previous work [18]. An important implication for this work is that we cannot assign Mg-content based on FIR peak shifts, even though this works very well in the mid-IR energy range [12,29].

Given the diversity of different Mg-containing phases that are known to form from aqueous syntheses [41,42], the use of IR spectral analyses to assess complex multi-phase products is useful. For example, elemental analyses do not add much useful information for phase identification since the likely products share many of the same elements. Furthermore, poor crystallinity makes constant PXRD lattice refinements challenging, especially since some of the likely Mg-containing phases have low-symmetry unit cells with many diffraction peaks (for example, the monoclinic hydromagnesite), as well as variable compositions (leading to diffraction peak shifts).

### FIR and OH- and H$_2$O-Containing Phases

In this work, our correlated analysis of MIR, FIR, and PXRD provides useful insights related to poorly crystallized carbonate phases that are hydrated and/or hydroxylated.

There is a precedent for tracking water peaks in the ACC synthesis literature [40,43], yet it is not routine to track the water-related spectral features in FIR data. In our case, this is an important consideration for interpreting our FIR spectra. In more general terms, the FIR spectra for water are widely studied, since rotational modes fall in this energy region. The FIR spectrum of liquid water is well-known and is broad. However, studies of other materials that involve structural water have used changes in these water peaks to assess changes in the water's environment, which can be triggered by changes to the solid [44]. Other studies [10] have followed the crystallization Mg-specific hydrated and hydroxylated phases using techniques other than IR (including PXRD, Raman spectroscopy, and thermogravimetric analyses).

Bringing this back to the data presented herein, we see that for samples prepared without EtOH (Figure 1e,f,g,h), there are dramatic decreases in the spectral signatures for water-related peaks (OH-stretching in the 3000–4000 cm$^{-1}$ range) after one year of aging, during which time the sample crystallizes into a mixture of dolomite and aragonite. In contrast, for samples prepared with EtOH, there is less change in the OH-stretch peak (3000–4000 cm$^{-1}$) over time, but the peak remains at a relatively high intensity even after 1 year; furthermore, the FIR data (Figure 3b) show a suite of peaks in the 300–600 cm$^{-1}$ range that are consistent with those reported for a range of different hydroxylated and hydrated carbonate mineral phases [20]. This assignment is also consistent with PXRD data (Figure 2b), which shows a strong yet unindexable peak near 7 degrees $2\theta$ that is characteristic of hydromagnesite phases with variable compositions (such as $Mg_5(CO_3)_4OH_2\cdot5H_2O$, ICSD PDF #29-0858 [28]).

Thus, a more complete explanation of the multiphase product emerges only once we incorporate all of these structural characterization details together (mid-IR, far-IR, and PXRD), as well as knowledge of the aqueous precipitation conditions.

## 4. Conclusions

We show clear spectral differences that allow us to track products that crystallize from Mg-containing amorphous calcium carbonate. The FIR data do not provide evidence that suggests whether the samples begin as proto-crystalline phases, or instead as inhomogeneous amorphous phases. However, they do assist in tracking OH-containing phases and those with structural water, phases that were not clearly identified from PXRD data. Taken altogether, this work outlines a generalizable strategy to integrate FIR spectral analyses with other structural characterization data to identify the crystallization products from amorphous precursors.

**Supplementary Materials:** The following are available online at https://www.mdpi.com/article/10.3390/min13010110/s1, Figure S1: Additional representative mid-IR spectra to show changes over time, or with mild heating.

**Author Contributions:** Conceptualization, methodology, writing—original draft preparation, review and editing: B.G. and K.M.P.; investigation and data curation: B.G.; supervision, project administration, and funding acquisition: K.M.P. All authors have read and agreed to the published version of the manuscript.

**Funding:** This research was funded by the Natural Science and Engineering Research Council of Canada (NSERC) grant number 2018-04888.

**Data Availability Statement:** The data presented in this study are available on request from the corresponding author.

**Acknowledgments:** J.B. Lin (X-ray diffraction at the Centre for Chemical Research and Training through Memorial University's CREAIT network) for access to characterization facilities and assistance with data analysis.

**Conflicts of Interest:** The authors declare no conflict of interest. The funders had no role in the design of the study; in the collection, analyses, or interpretation of data; in the writing of the manuscript, or in the decision to publish the results.

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
