# Peer review of "Tracking Amorphous Calcium Carbonate Crystallization Products with Far-Infrared Spectroscopy"

_minerals, doi:10.3390/min13010110_

Round 1
Reviewer 1 Report
It's not a surprise that FIR is not useful for identifying multi-phase Mg containing amorphous calcium carbonates, or proton-aragonite, port-dolomite, port-calcite etc. XRD and Raman spectroscopy would be the analytical techniques of choice to distinguish effectively between these phases. That said, I don't recall any published paper outlining the futility of applying FIR on these phases, so therefore, this work is worth publishing.
Author Response
No changes suggested by Reviewer 1.
Reviewer 2 Report
For a long time, it is widely accepted that biogenic carbonate minerals are formed via an ACC (amorphous calcium carbonate) precursor way. Nevertheless, the detailed crystallization process of these precursors is still elusive. In this respect, the manuscript is meaningful in that it provides an additional method, namely, Far-Infrared Spectroscopy (FIR), to track the ACC transformation process. It is well written with rich and reliable experimental data. In addition, the discussion section has been well developed.
However, there are two key questions about the aged products which need to be explained:
1 For the sample with EtOH aged for 1y, the author infers that it contains dolomite with minor aragonite, which is supported by the PXRD and MIR methods. However, the author concludes that it shows the features of calcite by the FIR method. The above contradiction is inconceivable and not well explained. It is well known that PXRD is definite in identifying a crystalline phase, and the same is true of the Infrared spectroscopy. Therefore, the author should provide more information, such as using the peak-fitting methods, to clear the above contradiction.
2 During ACC transformation, vaterite often occurs. Is it possible that a minor vaterite may exist in the samples with or without EtOH aged for 1y?
In addition, there are some typos needed to be corrected, such as:
Line 91-92: aging for a broader range of aging, from 3 days to 2 years.)
Line 126: For the aged sample made with EtOH (Figure 3d,
Line 149: This diversity of crystalline products are not surprising, based on the literature.
Round 2
Reviewer 2 Report
The authors have well explained my concerns and I recommend that the manuscript can be published in the present form.